# Instance Selection for GANs

**Terrance DeVries**
University of Guelph
Vector Institute

**Michal Drozdzal**
Facebook AI Research

**Graham W. Taylor**
University of Guelph
Vector Institute

## Abstract

Recent advances in Generative Adversarial Networks (GANs) have led to their widespread adoption for the purposes of generating high quality synthetic imagery. While capable of generating photo-realistic images, these models often produce unrealistic samples which fall outside of the data manifold. Several recently proposed techniques attempt to avoid spurious samples, either by rejecting them after generation, or by truncating the model's latent space. While effective, these methods are inefficient, as a large fraction of training time and model capacity are dedicated towards samples that will ultimately go unused. In this work we propose a novel approach to improve sample quality: altering the training dataset via instance selection before model training has taken place. By refining the empirical data distribution before training, we redirect model capacity towards high-density regions, which ultimately improves sample fidelity, lowers model capacity requirements, and significantly reduces training time. Code is available at `https://github.com/uoguelph-mlrg/instance_selection_for_gans`.

## 1 Introduction

Recent advances in Generative Adversarial Networks (GANs) have enabled these models to be considered a tool of choice for vision synthesis tasks that demand high fidelity outputs, such as image and video generation [6, 12], image editing [41], inpainting [35], and superresolution [32]. However, when sampling from a trained GAN model, outputs may be unrealistic just as often as they appear photo-realistic.

GANs fit a model to a data distribution with the help of a discriminator network. Low quality samples produced by these models are often attributed to poor modeling of the low-density regions of the data manifold [11]. The majority of current techniques attempt to eliminate low quality samples after the model is trained, either by changing the model distribution by truncating the latent space [2, 11] or by performing some form of rejection sampling using a trained discriminator to inform the rejection process [1, 5, 31]. Nevertheless, these methods are inefficient with respect to model capacity and training time, since much of the capacity and optimization efforts dedicated to representing the sparse regions of the data manifold are wasted.

In this paper, we analyze the use of instance selection [21] in the generative setting. We address the problem of uneven model sample quality before GAN model training has begun, rather than after it has finished. We note that dataset collection is a noisy process, and that many of the currently used datasets for generative model training and evaluation were not purposely created for this task. Thus, through a dataset curation step, we remove low density regions from the data manifold prior to model optimization and show that this *direct* dataset intervention (1) improves overall image sample quality in exchange for some reduction in diversity, (2) lowers model capacity requirements, and (3) reduces training time. To remove the sparsest parts of the image manifold, images are first projected into an embedding space of perceptually meaningful representations. A scoring function is then fit to asses the manifold density in the neighbourhood of each embedded data point in the dataset. Finally, data points with the lowest manifold density scores are removed from the dataset. In

our experiments, we evaluate a variety of image embeddings and scoring functions, observing that Inceptionv3 and Gaussian likelihood are well suited for the respective roles. Overall, we make the following contributions:

- We propose dataset curation via instance selection to improve the output quality of GANs.
- We show that the manifold density in the perceptual embedding space of a given dataset is predictive of GAN performance, and therefore a good scoring function for instance selection.
- We demonstrate the *model capacity savings* of instance selection by achieving state-of-the-art performance (in terms of FID) on $64 \times 64$ resolution ImageNet generation using a Self-Attention GAN with ½ the amount of trainable parameters of the current best model.
- We demonstrate *training time savings* by training a $128 \times 128$ resolution BigGAN on ImageNet in ¼ the time of the baseline, while also achieving superior performance across all image fidelity metrics.
- We exhibit the overall computational savings of instance selection by training a $256 \times 256$ resolution BigGAN on ImageNet with only 4 V100 GPUs in 11 days. Our model achieves better image fidelity than the baseline model while using ½ as many trainable parameters.

## 2 Related Work

Generative modelling of images is a very challenging problem due to the high dimensional nature of images and the complexity of the distributions they form. Several different approaches towards image generation have been proposed, with GANs currently the state-of-the-art in terms of image generation quality. In this work we will focus primarily on GANs, but other types of generative models might also benefit from instance selection prior to model fitting.

### 2.1 Sample Filtering in GANs

One way to improve the sample quality from GANs without making any changes to the architecture or optimization algorithm is by applying techniques which automatically filter out poor quality samples from a trained model. Discriminator Rejection Sampling (DRS) [1] accomplishes this by performing rejection sampling on the generator. This process is informed by the discriminator, which is reused to estimate density ratios between the real and generated image manifolds. Metropolis-Hastings GAN (MH-GAN) [31] builds on DRS by i) calibrating the discriminator to achieve more accurate density ratio estimates, and by ii) applying Markov chain Monte Carlo (MCMC) instead of rejection sampling for better performance on high dimensional data. Ding et al. [8] further improve density ratio estimates by fine-tuning a pretrained ImageNet classifier for the task. For more efficient sampling, Discriminator Driven Latent Sampling (DDLS) [5] iteratively updates samples in the latent space to push them closer to realistic outputs.

Instead of filtering samples after the GAN has been trained, some methods do so during the training procedure. Latent Optimisation for Generative Adversarial Networks (LOGAN) [33] optimizes latent samples each iteration at the cost of an additional forward and backward pass. Sinha et al. [27] demonstrate that gradients from low quality generated samples drive the model away from the nearest mode rather than towards it. As such, gradients from the worst samples each iteration during training may be ignored to improve generation quality.

Perhaps the most well known approach for increasing sample fidelity in GANs is the "truncation trick" [2, 11, 16]. The truncation trick is used in the popular models BigGAN [2] and StyleGAN [11, 12] to improve image quality by manipulating the latent distribution. The original truncation trick as used by BigGAN consists of replacing the latent distribution with a truncated distribution during inference, such that any latent sample that falls outside of some acceptable range is resampled. StyleGAN uses a similar strategy by interpolating samples towards the mean of the latent space instead of resampling them. By moving samples closer to the interior regions of the latent space, sample diversity can effectively be traded for visual fidelity. Our instance selection technique has an effect similar to the truncation trick, but with the added benefit of also reducing model capacity and training time requirements.

## 2.2 Instance Selection

Instance selection is a data preprocessing technique commonly used in the classification setting to select a subset of data from a larger collection [21]. In general, instance selection methods either attempt to reduce the size of the dataset to a more manageable size while retaining informative data points, or try to clean the dataset by eliminating noisy data points. Though commonly used in the setting of big data, instance selection has received little attention from the generative modelling community. Nuha et al. [20] explore the impact of reducing the size of the training set when training GANs. However, they select data points randomly, and no significant improvement in performance is observed from the removal of data. Core-set selection has been shown to be useful for improving GAN performance when training with small mini-batches, but it ultimately does not improve image fidelity over large mini-batch training [26]. Whereas core-set selection attempts to select mini-batches that mimic the distribution of the original dataset, our proposed technique purposefully redefines the target distribution so as to maximize the density of the data manifold.

## 3 Instance Selection for GANs

In the context of generative modeling, our motivation is to automatically remove the sparsest regions of the data manifold, specifically those parts that GANs struggle to capture. To do so, we define an image embedding function $F$ and a scoring function $H$.

**Embedding function** $F$ projects images into an embedding space. More precisely, given a dataset of images $\mathcal{X}$, the dataset of embedded images $\mathcal{Z}$ is obtained by applying the embedding function $\mathbf{z} = F(\mathbf{x})$ to each data point $\mathbf{x} \in \mathcal{X}$. For the task of image generation we suggest using perceptually aligned embedding functions [37], such as the feature space of a pretrained image classifier.

**Scoring function** $H$ is used to to assess the manifold density in a neighbourhood around each embedded data point $\mathbf{z}$. In our experiments, we compare three choices of scoring function: log likelihood under a standard Gaussian model, log likelihood under a Probabilistic Principal Component Analysis (PPCA) [29] model, and distance to the $K^{\text{th}}$ nearest neighbour (KNN Distance). We select Gaussian and PPCA as simple, well known density models. KNN Distance has previously been used as a measure of local manifold density in classical instance selection [3], and has been shown to be useful for defining non-linear image manifolds [14, 19].

The Gaussian model is fit to the embedded dataset by computing the empirical mean $\boldsymbol{\mu}$ and the sample covariance $\boldsymbol{\Sigma}$ of $\mathcal{Z}$. The score of each embedded image $\mathbf{z}$ is computed as follows:

$$H_{\text{Gaussian}}(\mathbf{z}) = -\frac{1}{2}[\ln(|\boldsymbol{\Sigma}|) + (\mathbf{z} - \boldsymbol{\mu})^T \boldsymbol{\Sigma}^{-1}(\mathbf{z} - \boldsymbol{\mu}) + d\ln(2\pi)], \tag{1}$$

where $d$ is the dimension of $\mathbf{z}$.

PPCA is fit to the embedded dataset using any standard PPCA solver [22]. We set the number of principal components such that $95\%$ of the variance in the data is preserved. Embedded images are scored as follows:

$$H_{\text{PPCA}}(\mathbf{z}) = -\frac{1}{2}[\ln(|\mathbf{C}|) + \text{Tr}((\mathbf{z} - \boldsymbol{\mu})^T \mathbf{C}^{-1}(\mathbf{z} - \boldsymbol{\mu})) + d\ln(2\pi)], \quad \mathbf{C} = \mathbf{W}\mathbf{W}^T + \sigma^2\mathbf{I}, \tag{2}$$

where $\mathbf{W}$ is the fit model weight matrix, $\boldsymbol{\mu}$ is the empirical mean of $\mathcal{Z}$, $\sigma$ is the residual variance, $\mathbf{I}$ is the identity matrix, and $d$ is the dimension of $\mathbf{z}$.

KNN Distance is used to score data points by calculating the Euclidean distance between $\mathbf{z}$ and $\mathcal{Z} \setminus \{\mathbf{z}\}$, then returning the distance to the $K_{th}$ nearest element. To convert to a score, we make the resulting distance negative, such that smaller distances return larger values. Formally, we can evaluate:

$$H_{\text{KNN}}(\mathbf{z}, K, \mathcal{Z}) = -\min_K \left\{ ||\mathbf{z} - \mathbf{z}_i||_2 \quad : \quad \mathbf{z}_i \in \mathcal{Z} \setminus \{\mathbf{z}\} \right\}, \tag{3}$$

where $\min_K$ is defined as the $K_{th}$ smallest value in a set. In our experiments we set $K = 5$.

To perform **instance selection**, we compute scores $H(F(\mathbf{x}))$ for each data point and keep all data points with scores above some threshold $\psi$. For convenience, we often set $\psi$ to be equal to some percentile of the scores, such that we preserve the top $N\%$ of the best scoring data points. Thus,

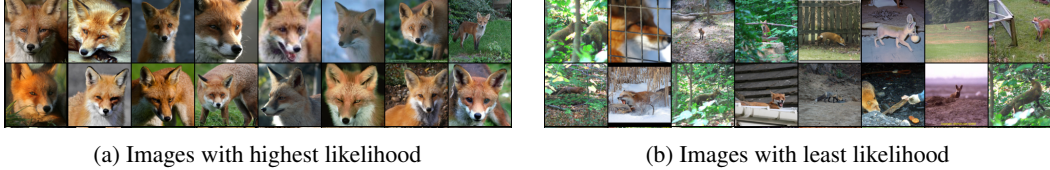

<div align="center">(a) Images with highest likelihood          (b) Images with least likelihood</div>

Figure 1: Examples of the (a) most and (b) least likely resized images of red foxes from the ImageNet dataset, as determined by a Gaussian model fit on images in an Inceptionv3 embedding space. High likelihood images share a similar visual structure, while low likelihood samples are more varied.

given an initial training set consisting of data points $\mathbf{x} \in \mathcal{X}$ we construct our reduced training set $\mathcal{X}'$ by computing:

$$\mathcal{X}' = \{\mathbf{x} \in \mathcal{X} \quad \text{s.t.} \quad H(F(\mathbf{x})) > \psi\}. \tag{4}$$

To illustrate why removing data points from the training set might be a good idea, we look at the most and least likely images from the red fox class of ImageNet (Figure 1). Likelihood is determined by a Gaussian model fit on feature embeddings from a pretrained Inceptionv3 classifier. We notice a stark contrast between the content of the images. The most likely images (a) are similarly cropped around the fox's face, while the least likely images (b) have many odd viewpoints and often suffer from occlusion. It is logical to imagine how a generative model trained on these unusual instances may try to generate samples that mimic such conditions, resulting in undesirable outputs.

## 4    Experiments

In this section we review evaluation metrics, motivate selecting instances based on manifold density, and then analyze the impact of applying instance selection to GAN training.

### 4.1    Evaluation Metrics

We use a variety of evaluation metrics to diagnose the effect that training with instance selection has on the learned distribution, including: (1) Inception Score (IS) [24], (2) Fréchet Inception Distance (FID) [10], (3) Precision and Recall (P&R) [14], and (4) Density and Coverage (D&C) [19]. In all cases where a reference distribution is required we use *the original training distribution*. Using the distribution produced after instance selection would unfairly favour the evaluation of instance selection, since the reference distribution could be changed to one that is trivially easy to generate. A detailed description of each evaluation metric is provided in the supplementary material (§A).

When calculating FID we follow Brock et al. [2] in using all images in the training set to estimate the reference distribution, and sampling 50 k images to make up the generated distribution. For P&R and D&C we use an Inceptionv3 embedding.[1] $N$ and $M$ are set to 10 k samples for both the reference and generated distributions, and $K$ is set equal to 5 as recommended by Naeem et al. [19].

### 4.2    Relationship Between Dataset Manifold Density and GAN Performance

An image manifold is more accurately defined in regions where many data points are in close proximity to each other [14]. Since GANs attempt to reproduce an image manifold based on data points from a given dataset, we suspect that they should perform better on datasets with well-defined manifolds (i.e. no sparse manifold regions). To verify this hypothesis, we use the ImageNet[2] dataset [7] and treat each of the 1000 classes as a separate dataset. Ideally, we would fit a separate GAN on each class to obtain a ground truth measure of performance, but this is very computationally expensive. Instead, we use a single class-conditional BigGAN from [2] that has been pretrained on ImageNet at $128 \times 128$ resolution. For each class, we sample 700 real images from the dataset, and generate 700 class-conditioned samples with the BigGAN. To measure the density for each class manifold we compare three different methods: Gaussian likelihood, Probabilistic Principal Component Analysis

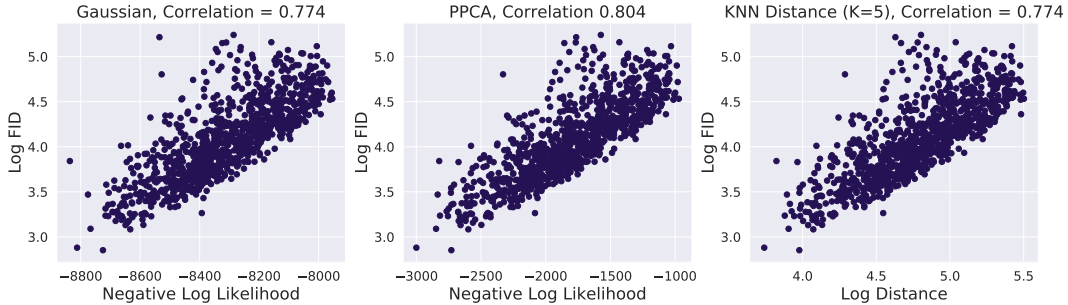

Figure 2: Correlation between manifold density estimates and FID for each class in the ImageNet dataset. Lower values on the x-axis indicate a more dense dataset manifold. Lower values on the y-axis indicate better quality generated samples.

(PPCA) likelihood, and distance to the $K^{\text{th}}$ neighbour (KNN Distance) (§3). Images are projected into the feature space of an Inceptionv3 model, and a manifold density score is computed on the features using one of our scoring functions. As an indicator of the true GAN output quality we compute FID between the real and generated distributions for each class.

We observe a strong correlation between each of the manifold density measures and GAN output quality (Figure 2). This correlation confirms our hypothesis, suggesting that dataset manifold density is an important factor for achieving high quality generated samples with GANs.

## 4.3 Embedding and Scoring Function

Having established that dataset manifold density is correlated with GAN performance, we explore artificially increasing the overall density of the training set by removing data points that lie in low density regions of the data manifold. To this end, we train several Self-Attention GANs (SAGAN) [36] on ImageNet at $64 \times 64$ resolution. Each model is trained on a different $50\%$ subset of ImageNet, as chosen by instance selection using different embedding and scoring functions as described in §3. Instance selection is applied per-class. We use the default settings for SAGAN, except that we use a batch size of 128 instead of 256, apply the self-attention module at $32 \times 32$ resolution instead of $64 \times 64$, and reduce the number of channels in each layer by half in order to reduce the computational cost of our initial exploratory experiments. All models are trained for 200k iterations. The results of these experiments are shown in Table 1. For reference, we include scores achieved by real (i.e. not generated) data in Table 5 in the supplementary material.

Table 1: Comparison of embedding and scoring functions on $64 \times 64$ ImageNet image generation task. All tests train a SAGAN model for 200k iterations. Models trained with instance selection significantly outperform models trained without instance selection, despite training on a fraction of the available data. RR is the retention ratio (percentage of dataset trained on). Best results in bold.

| Instance Selection | RR (%) | Embedding | Pretraining | IS ↑ | FID ↓ | P ↑ | R ↑ | D ↑ | C ↑ |
|---|---|---|---|---|---|---|---|---|---|
| None | 100 | - | - | 15.4 | 21.4 | 0.66 | **0.62** | 0.64 | 0.64 |
| Uniform | 50 | - | - | 15.5 | 22.8 | 0.65 | **0.62** | 0.65 | 0.65 |
| Gaussian | 50 | Inceptionv3 | ImageNet | **25.7** | **12.6** | **0.77** | 0.59 | **0.97** | **0.83** |
| PPCA | 50 | Inceptionv3 | ImageNet | 25.5 | 13.2 | 0.76 | 0.58 | **0.97** | 0.82 |
| KNN Dist | 50 | Inceptionv3 | ImageNet | 25.4 | 13.1 | 0.76 | 0.58 | **0.97** | 0.82 |
| Gaussian | 50 | Inceptionv3 | Random init | 15.5 | 21.9 | 0.66 | 0.61 | 0.68 | 0.65 |
| Gaussian | 50 | ResNet-50 | Places365 | 20.6 | 16.5 | 0.74 | 0.59 | 0.88 | 0.76 |
| Gaussian | 50 | ResNet-50 | SwAV | 20.3 | 16.7 | 0.74 | 0.57 | 0.89 | 0.76 |
| Gaussian | 50 | ResNet-50 | ImageNet | 22.0 | 14.6 | 0.76 | 0.59 | 0.92 | 0.79 |
| Gaussian | 50 | ResNeXt-101 | Instagram 1B | 24.1 | 14.1 | 0.73 | 0.61 | 0.86 | 0.80 |

All runs utilizing instance selection significantly outperform the baseline model trained on the full dataset, despite only having access to half as much training data (Table 1). We observe a large increase in image fidelity, as indicated by the improvements in Inception Score, Precision, and Density, and a slight drop in overall diversity, as measured by Recall. Coverage, which measures realism-constrained diversity, benefits greatly from the more realistic samples and thus sees an increase, despite the reduction in overall diversity. Since the increase in image quality is much greater than the decrease in diversity, FID also improves. To verify that the gains are not simply caused by the reduction in dataset size we train a model on a 50% subset that was uniform-randomly sampled from the full dataset. Here, we observe little change in performance compared to the baseline, indicating that performance improvements are indeed due to careful selection of training data, rather than the reduction of dataset size.

We find that all three candidate scoring functions: Gaussian likelihood, PPCA likelihood, and KNN distance, significantly outperform the full dataset baseline. Gaussian likelihood slightly outperforms the alternatives, so we use it as the scoring function in the remainder of our experiments.

To understand the importance of the embedding function, we compare several different model embeddings that have been trained on different datasets: Inceptionv3 [28] trained on ImageNet, ResNet50 [9] trained on Places365 [40], ImageNet, and with SwAV unsupervised pretraining [4], and ResNeXt-101 32x8d [34] trained with weak supervision on Instagram 1B [15]. We also compare a randomly initialized Inceptionv3 with no pretraining as a random embedding. For all architectures, features are extracted after the global average pooling layer. We find that all feature embeddings improve performance over the full dataset baseline except for the randomly initialized network. These results suggest that an embedding function that is well aligned with the target domain is required in order for instance selection to be effective. The ImageNet pretrained Inceptionv3 embedding performs best overall, and was chosen as the embedding function for the rest of our experiments. We note that using an Inceptionv3 embedding both in instance selection and in the evaluation metrics may yield some non-negligible advantage in evaluation, since selected instances are those that the network prefers.

## 4.4 Retention Ratio

An important consideration when performing instance selection is determining what proportion of the original dataset to keep, a hyperparameter which we call retention ratio. To investigate the impact of the retention ratio on training, we train ten SAGANs on ImageNet, each retaining different amounts of the original dataset in $10\%$ intervals. GAN hyperparameters are the same as in §4.3, except that we extend training until 500k iterations in order to observe model behaviours over a longer training window. Results are shown in Figure 3 and Table 6 in the supplementary material.

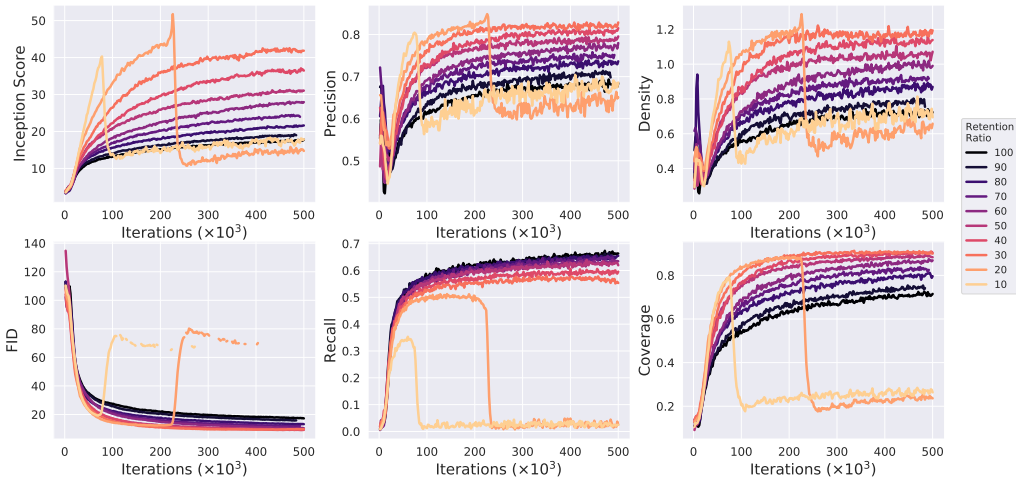

Figure 3: SAGAN trained on $64 \times 64$ ImageNet, with instance selection used to reduced the dataset by varying amounts. Retention ratio = 100 indicates a model trained on the full dataset (i.e. no instance selection). The application of instance selection boosts overall performance significantly.

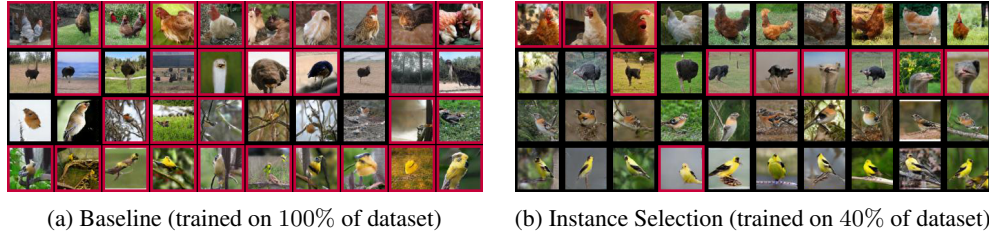

| (a) Baseline (trained on $100\%$ of dataset) | (b) Instance Selection (trained on $40\%$ of dataset) |

Figure 4: Samples of bird classes from SAGAN trained on $64 \times 64$ ImageNet. Each row is conditioned on a different class. Red borders indicate misclassification by a row-specific pretrained Inceptionv3 classifier. Instance selection (b) significantly improves sample fidelity and class consistency compared to the baseline (a).

As larger portions of the original dataset are removed we see consistent improvements in image fidelity (increasing Inception Score, Precision, and Density) and reductions in sample diversity (decreasing Recall). Interestingly, metrics which take into account both realism and diversity (FID and Coverage) continue to see gains until roughly $70\%$ of the dataset has been removed, at which point they begin to decrease. This behaviour suggests that, given the ability of current state-of-the-art models to learn from limited data, sample fidelity is valued much more than diversity. When too much of the dataset is removed some models collapse prematurely, likely due to the discriminator quickly overfitting the small training set. It is expected that applying data augmentation could resolve this issue [13, 38]. To further improve image fidelity, instance selection could be combined with the truncation trick (§E).

Our best performing SAGAN model in terms of FID was trained on only $40\%$ of the ImageNet dataset, yet *outperforms* FQ-BigGAN [39], the current state-of-the-art model for the task of $64 \times 64$ ImageNet generation. Despite using $2\times$ less parameters and a $4\times$ smaller batch size, our SAGAN achieves a better FID (9.07 vs. 9.76). As indicated by these scores and the errors made by a pretrained classifier, samples from our instance selection model are significantly more recognizable than those from the baseline model trained on the full dataset (Figure 4).

### 4.5    128 × 128 ImageNet

To examine the impact of instance selection on the training time of large-scale models, we train two BigGAN models on $128 \times 128$ ImageNet[3]. Our baseline model uses the default hyperparameters from BigGAN [2], with the exception that we reduce the channel multiplier from 96 to 64 (i.e. half of the capacity) and only use a single discriminator update instead of two for faster training. Our instance selection model uses the same settings as the baseline, but is trained on $50\%$ of the dataset. Although large batch sizes are critical for achieving good performance with the baseline BigGAN [2], we found them to degrade performance when combined with instance selection. Therefore, we reduce the batch size from BigGAN's default of 2048 to 256 for the instance selection model. Both models are trained on 8 NVIDIA V100 GPUs with 16GB of RAM, using gradient accumulation to achieve the necessary batch sizes.

Despite using a much smaller batch size, our model trained with instance selection outperforms the baseline in all metrics except for Recall (Table 2), as expected due to the diversity/fidelity trade-off. The instance selection model trains significantly faster than the baseline, requiring less than four days while the baseline requires more than two weeks.

Table 2: Performance of models on the $128 \times 128$ ImageNet image generation task. Both models use a channel multiplier of 64 and a single discriminator update per generator update. The baseline model uses a batch size of 2048, while the instance selection model uses a batch size of 256.

| Model | IS ↑ | FID ↓ | P ↑ | R ↑ | D ↑ | C ↑ | Time ↓ | Hardware |
|---|---|---|---|---|---|---|---|---|
| BigGAN | 68.8 | 11.5 | 0.76 | **0.66** | 0.9 | 0.84 | 14.8 days | 8 V100 |
| BigGAN + Inst. Sel. | **114.3** | **9.6** | **0.88** | 0.50 | **1.34** | **0.90** | **3.7 days** | 8 V100 |

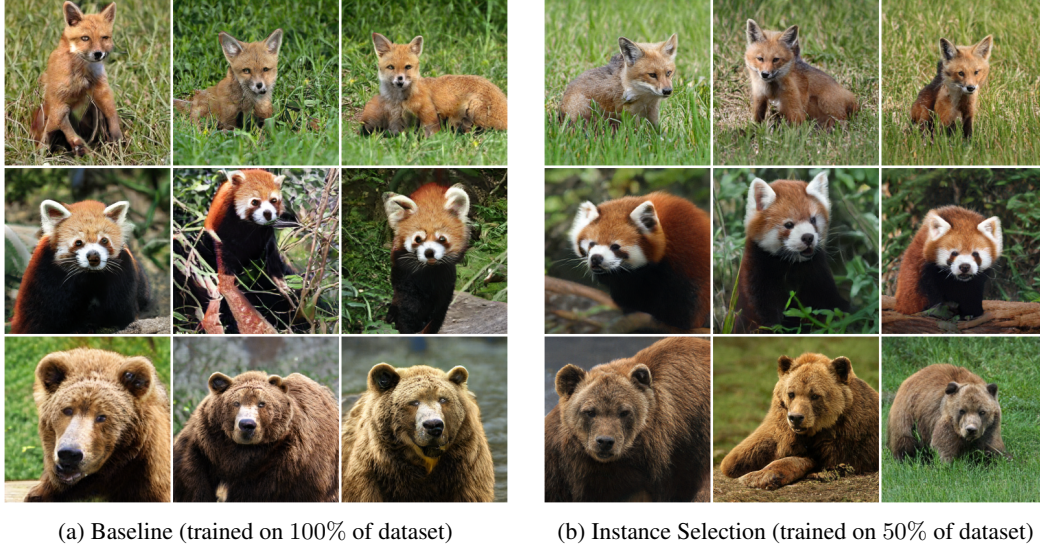

<div align="center">

(a) Baseline (trained on $100\%$ of dataset)    (b) Instance Selection (trained on $50\%$ of dataset)

</div>

Figure 5: Samples from BigGAN trained on $256 \times 256$ ImageNet, with the truncation trick. Samples are selected to demonstrate the highest quality outputs for each model. The baseline model (a) struggles to produce convincing facial details, which the instance selection model (b) successfully achieves. Zoom in for best viewing.

## 4.6  256 ×256 ImageNet

To further demonstrate instance selection we train a BigGAN on ImageNet at $256 \times 256$ resolution using 4 V100s with 32GB of RAM each. Since training a baseline model without instance selection with the same hardware setup would take an excessively long time (1-2 months), we instead compare to the $256 \times 256$ BigGAN from Brock et al. [2] using the official pretrained weights[4]. Compared to this baseline, our model uses half the capacity (channel multiplier reduced from 96 to 64), 8× smaller batch size (from 2048 to 256), and applies the self-attention block in the generator at a resolution of $64 \times 64$ instead of $128 \times 128$. The retention ratio for instance selection is set to $50\%$. Similar to the baseline, we use two discriminator update steps per generator update for this experiment. Quantitative results are presented in Table 3, and samples are shown in Figure 5 and §G in the supplementary material.

Our instance selection model trains in less than 11 days, and uses approximately one order of magnitude less multiply-accumulate operations (MACS) than the baseline throughout the duration of training. Despite having half as much capacity, our model outperforms the baseline in all image fidelity focused metrics (Inception Score, Precision, and Density), and achieves comparable performance on metrics that jointly consider image quality and diversity (FID and Coverage). As expected, the better image quality comes at the cost of overall sample diversity (indicated by Recall). To our knowledge, this is the first time photorealistic generation of $256 \times 256$ ImageNet images has been achieved without the use of specialized hardware (i.e. hundreds of TPUs).

Table 3: Performance of models for $256 \times 256$ ImageNet image generation. The instance selection model uses half as many parameters as the baseline model. All metrics are computed using PyTorch Inceptionv3 embeddings, and may therefore differ from numbers computed with TensorFlow.

| Model | IS ↑ | FID ↓ | P ↑ | R ↑ | D ↑ | C ↑ | Time | Hardware |
|---|---|---|---|---|---|---|---|---|
| BigGAN | 135.4 | **9.8** | 0.86 | **0.70** | 1.18 | 0.92 | 1-2 days | 256 TPUv3 |
| BigGAN + Inst. Sel. | **165.3** | 10.6 | **0.91** | 0.52 | **1.48** | **0.93** | 10.7 days | 4 V100 |

# 5    Instance Selection in Practice

As the experiments have shown, instance selection stands as a useful tool for trading away sample diversity in exchange for improvements in image fidelity, faster training, and lower model capacity requirements. We believe that this trade-off is a worthwhile hyperparameter to tune in consideration of the available compute budget, just as it is common practice to adjust model capacity or batch size to fit within the memory constraints of the available hardware.

The control over the diversity/fidelity trade-off afforded by instance selection also yields a tool that can be used to better understand the behaviour and limitations of existing evaluation metrics. For instance, in some cases when applying instance selection, we observed that certain diversity-sensitive metrics (such as FID and Coverage) improved, even though the diversity of the training set had been significantly reduced. We leave it for future work to determine whether this is a limitation of these metrics, or a behaviour that should be expected.

Finally, instance selection can be used to automatically curate new datasets for the task of image generation. Existing datasets that are designed for image synthesis often use manual filtering and hand-crafted cropping and alignment tools to increase the dataset manifold density [11]. As an alternative to these time-intensive procedures, instance selection provides a generic solution that can quickly be applied to any uncurated set of images.

# 6    Conclusion

Folk wisdom suggests *more data is better*, however, it is known that areas of the data manifold that are sparsely represented pose a challenge to current GANs [11]. To directly address this challenge we introduce a new tool: dataset curation via instance selection. Our motivation is to remove sparse regions of the data manifold before training, acknowledging that they will ultimately be poorly represented by the GAN, and therefore, that attempting to capture them is an inefficient use of model capacity. Moreover, popular post-processing methods such as rejection sampling or latent space truncation will likely ignore these regions as represented by the model. There are multiple benefits of taking the instance selection approach: (1) We improve sample fidelity across a variety of metrics compared to training on uncurated data; (2) We demonstrate that reallocating model capacity to denser regions of the data manifold leads to efficiency gains, meaning that we can achieve SOTA quality with smaller-capacity models trained in far less time. To our knowledge, instance selection has not yet been formally analyzed in the generative setting. However, we argue that it is more important here than in supervised learning because of the absence of an annotation phase where humans often perform some kind of formal or informal curation.

We have only considered the setting where curation is performed up-front, prior to training. However, our results suggest that dynamic curation, including curriculum learning informed by the kinds of perceptually aligned embeddings we consider here, is an interesting direction for future work.

## Broader Impact

The application of instance selection to the task of image generative modelling brings with it several benefits. Gains in image generation quality are an obvious improvement, but perhaps more impactful to the broader community are the reductions in model capacity and training time that are afforded. Reducing the computational barrier to entry for training large-scale generative models provides many individuals, including students, AI artists, and ML enthusiasts, with access to models that are otherwise restricted to only the most well resourced labs. In addition to greater accessibility, lowering the computational requirements for training large-scale generative models also reduces associated energy costs and $CO_2$ emissions associated with the training process.

One side effect of our instance selection method is that, by nature of design, generated results are more likely to reflect the content that makes up the majority of the training set. As such, dataset bias is amplified as instances that are poorly represented in the dataset may be completely ignored. However, this limitation can be addressed by properly balancing the training set before instance selection is applied or alternatively, ensuring a more diverse & inclusive data collection effort to begin with.

As with any form of generative model, there is some potential for misuse. A common example is "deepfakes", where a generative model is used to manipulate images or videos well enough that humans cannot distinguish real from fake. While often used to create humorous videos in which actors' faces are swapped, deepfakes also have the potential for more nefarious uses, such as for blackmail or spreading misinformation. Fortunately, much recent effort has been dedicated towards automatic detection of these false images [30]. These techniques attempt to find manipulated media by detecting inconsistencies, such as in the synchronization of lip movement and speech audio, or generation artifacts, such as missing reflections or other minute details.

## Acknowledgments and Disclosure of Funding

The authors would like to thank Colin Brennan, with whom discussions about dataset learnability kicked off this project, and Brendan Duke, for being a constant sounding board. Resources used in preparing this research were provided to GWT and TD, in part, by NSERC, the Canada Foundation for Innovation, the Province of Ontario, the Government of Canada through CIFAR, Compute Canada, and companies sponsoring the Vector Institute: `http://www.vectorinstitute.ai/#partners`.

## Footnotes

[1]We use the PyTorch pretrained Inceptionv3 embedding for all metrics.

[2]Use of ImageNet is only for noncommercial, research purposes, and not for training networks deployed in production or for other commercial uses.

[3]We use the official BigGAN implementation from `https://github.com/ajbrock/BigGAN-PyTorch`.

[4]Pretrained BigGAN weights from `https://colab.research.google.com/github/tensorflow/hub/blob/master/examples/colab/biggan_generation_with_tf_hub.ipynb`

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
