[Supplementary Material]

# A   Detailed Description of Evaluation Metrics

We use a variety of evaluation metrics to diagnose the effect that training with instance selection has on the learned distribution. In all cases where a reference distribution is required we use *the original training distribution*, and not the distribution produced after instance selection. Doing so would unfairly favour the evaluation of instance selection, since the reference distribution could be changed to one that is trivially easy to generate.

**Inception Score (IS)** [24] evaluates samples by extracting class probabilities from an ImageNet pretrained Inceptionv3 classifier and measuring the distribution of outputs over all samples. The Inception Score is maximized when a model produces highly recognizable outputs for each of the ImageNet classes. One of the major limitations of the Inception Score is its insensitivity to mode collapse within each class. A model that produces a single high quality image for each category can still achieve a good score.

**Fréchet Inception Distance (FID)** [10] measures the distance between a generated distribution and a reference distribution, as approximated by a Gaussian fit to samples projected into the feature space of a pretrained Inceptionv3 model. FID has been shown to correlate well with image quality, and is capable of detecting mode collapse and mode adding. However, FID does not differentiate between fidelity and diversity. As such, it is difficult to assess whether a model has achieved a good FID score based on good mode coverage, or because it produces high quality samples.

**Precision and Recall (P&R)** [14] were designed to address the limitations of FID by providing separate metrics to evaluate fidelity and diversity. To calculate P&R, image manifolds are created by first embedding each image in a given distribution into the feature space of a pretrained classifier. A radius is then extended from each data point to its $K^{\text{th}}$ nearest neighbour to form a hypersphere, and the union of all hyperspheres represents the image manifold. Precision is described as the percentage of generated samples that fall within the manifold of real images. Recall is described as the percentage of real images which fall within the manifold of generated samples. A limitation of P&R is that they are susceptible to outliers, both in the reference and generated distributions [19]. Outliers artificially inflate the size of the image manifolds, increasing the rate at which samples fall into those manifolds. Thus, a dataset or model that produces many outliers may achieve scores that are better than the quality of the samples would indicate.

**Density and Coverage (D&C)** [19] have recently been proposed as robust alternatives to Precision and Recall. Density can be seen as an extension of Precision which measures how many real image manifolds a generated sample falls within on average. Coverage is described as the percentage of real images that have a generated sample fall within their manifold.

**Classification Accuracy Score (CAS)** [23, 25] was introduced for evaluating the usefulness of conditional generative models for augmenting downstream tasks such as image classification. To compute CAS, generated samples are used to train a classifier, which is then used to classify real data from a test set. Generally, it is observed that models with greater sample diversity achieve higher CAS, with image fidelity being of less importance. We do not evaluate CAS for the majority of our experiments as it is very computationally expensive to compute, but we do report it in § B, Table 4 for our 128 × 128 ImageNet BigGAN experiments as a reference for how instance selection affects CAS.

# B   Additional Evaluation Metrics - Classification Accuracy Score (CAS)

We compute CAS by training a ResNet50 on samples from each of our 128 × 128 BigGAN models using the standard ImageNet pipeline from PyTorch[5]. We find that the model trained without instance selection achieves the best CAS, which is expected given that this model also produces more diverse samples (as measured by Recall). Interestingly, CAS for the BigGAN trained with instance selection drops by less than 1%, despite it only having seen 50% of the ImageNet training set. This result might suggest that neither of the models evaluated does a good job at generating recognizable outliers from the ImageNet training set.

Table 4: CAS for BigGAN trained with and without instance selection. Following [23], both models use a truncation ratio of 1.5 when generating samples for increased diversity.

| Training Set | Resolution | Top-5 Accuracy | Top-1 Accuracy |
|---|---|---|---|
| BigGAN | $128 \times 128$ | **18.73** | **9.21** |
| BigGAN + 50% inst. sel. | $128 \times 128$ | 17.94 | 8.42 |

## C   Scores of Evaluation Metrics on Real Data

For each evaluation metric we compute scores on real data (Table 5) as a reference for comparison with the values produced by generative models. These values can be thought of as the scores which would be achieved by a generative model that perfectly captures the target distribution. Metrics are evaluated on the ImageNet validation set, using all 50k data points for IS and FID and 10k randomly selected data points for P&R and D&C. Note that it is possible for generative models to surpass the scores of real data for metrics that focus on image fidelity, such as IS, P, and D, but these models often have proportionally lower diversity scores.

Table 5: Scores of real data from the ImageNet validation set for all evaluation metrics.

| Resolution | IS ↑ | FID ↓ | P ↑ | R ↑ | D ↑ | C ↑ |
|---|---|---|---|---|---|---|
| $64 \times 64$ | 59.1 | 1.0 | 0.79 | 0.79 | 0.99 | 0.96 |
| $128 \times 128$ | 148.2 | 1.2 | 0.84 | 0.82 | 1.01 | 0.96 |
| $256 \times 256$ | 225.9 | 1.4 | 0.85 | 0.83 | 1.01 | 0.96 |

## D   Retention Ratio Experiment Numerical Results

In Table 6 we include numerical results for the retention ratio experiments conducted in §4.4. These values accompany the plots in Figure 3. We also report the performance of BigGAN and FQ-BigGAN from [39] for comparison.

Table 6: Performance of models trained on $64 \times 64$ resolution ImageNet. A retention ratio of less than 100 indicates that instance selection is used. Best results in bold.

| Model | Params (M) | Batch Size | Retention Ratio (%) | IS ↑ | FID ↓ | P ↑ | R ↑ | D ↑ | C ↑ |
|---|---|---|---|---|---|---|---|---|---|
| BigGAN | 52.54 | 512 | 100 | 25.43 | 10.55 | - | - | - | - |
| FQ-BigGAN | 52.54 | 512 | 100 | 25.96 | 9.67 | - | - | - | - |
| SAGAN | 23.64 | 128 | 100 | 17.77 | 17.23 | 0.68 | **0.66** | 0.72 | 0.71 |
| | | | 90 | 18.98 | 15.85 | 0.70 | **0.66** | 0.75 | 0.74 |
| | | | 80 | 21.62 | 13.17 | 0.74 | 0.65 | 0.87 | 0.79 |
| | | | 70 | 23.95 | 11.98 | 0.75 | 0.64 | 0.92 | 0.82 |
| | | | 60 | 27.95 | 10.35 | 0.78 | 0.63 | 0.99 | 0.87 |
| | | | 50 | 31.04 | 9.63 | 0.79 | 0.62 | 1.07 | 0.88 |
| | | | 40 | 37.10 | **9.07** | 0.81 | 0.60 | 1.12 | **0.90** |
| | | | 30 | 41.85 | 9.75 | **0.83** | 0.55 | **1.19** | **0.90** |
| | | | 20 | **43.30** | 12.36 | 0.82 | 0.49 | 1.17 | 0.88 |
| | | | 10 | 37.16 | 19.24 | 0.79 | 0.33 | 1.07 | 0.78 |

# E Complementarity of Instance Selection and Truncation

The truncation trick is a simple and popular technique which is used to increase the visual fidelity of samples from a GAN at the expense of reduced diversity [2]. This trade-off is achieved by biasing latent samples towards the interior regions of the latent distribution, either by truncating the distribution, or by interpolating latent samples towards the mean [11, 14].

To examine the compatability between the truncation trick and instance selection, we truncate latent vectors of the models trained in §4.4, varying the truncation threshold from 1.0 to 0.1 (Figure 6). We observe that combining both techniques results in a greater improvement in visual fidelity than either method applied in isolation. We anticipate that other post-hoc filtering methods could also see complimentary benefits when combined with instance selection, such as DRS, MH-GAN, and DDLS.

Figure 6: Truncation trick applied to models trained with instance selection for truncation thresholds 1 to 0.1. The base models (threshold = 1) are marked with a ●. Up and to the right is best.

# F Insights for Applying Instance Selection to GANs

We found that, while instance selection could be used to achieve significant gains in model performance, some changes to other hyperparameters were necessary in order to ensure training stability. Here we detail some techniques that we found to work well in our experiments.

- **Reduce batch size** - Contrary to evidence from BigGAN [2] suggesting that larger batch sizes improve GAN performance, we found batch sizes larger than 256 to degrade performance when training with instance selection. We speculate that because we have simplified the training distribution by removing the difficult examples, the discriminator overfits the training set much faster. We posit that the smaller batch size could be acting as a form of regularization by reducing the accuracy of the gradients, thereby allowing the generator to train for longer before the discriminator overfits the training set and the model collapses.

- **Reduce model capacity** - Since the complexity of the training set is reduced when applying instance selection, we found it necessary in some cases to also reduce model capacity. Training models with too much capacity lead to early collapse, also likely caused by the discriminator quickly overfitting the training set. We note that with proper regularization, models trained with instance selection could still benefit from more capacity.

- **Apply additional regularization** - We have not experimented much with applying GAN regularization methods to our models, but think that it could be important for combating the aforementioned discriminator overfitting problem. Applying techniques such as R1 regularization [18] or recently proposed GAN data augmentation [13, 38] could allow for instance selection to be combined with the benefits of larger batch sizes and model capacity. We leave this investigation for future work.

# G   Sample Sheets

We generate several different sample visualizations in order to better understand the impact that instance selection has on GAN behaviour.

In Figure 7 we showcase some photorealistic samples generated by a $256 \times 256$ BigGAN model trained with instance selection.

In Figure 8 we compare randomly selected samples from the official pretrained $256 \times 256$ BigGAN (Figure 8a) with random samples from our $256 \times 256$ BigGAN trained with 50% instance selection (Figure 8b). Samples from the instance selection model appear more realistic on average.

To better understand how instance selection affects sample diversity, we visualize image manifolds of different datasets and models by organizing images in 2D using UMAP [17] (Figure 9). We only plot a single class so that we can see variations across the image manifold in greater detail than if multiple classes were plotted simultaneously. All image samples share the same 2D embedding, such that manifolds are comparable between datasets and models. We observe that even though instance selection has removed 50% of the images from the original dataset (Figure 9a), it still retains coverage over most of the original image manifold (Figure 9b). Only images containing extreme viewpoints are omitted. The GANs trained on the original and reduced datasets both cover less of the image manifold than their respective source datasets. While the baseline GAN (Figure 9c) covers more of the image manifold than the GAN trained with instance selection (Figure 9d), samples from these extra regions often appear less realistic.

Figure 7: Photorealistic samples from BigGAN trained on $256 \times 256$ ImageNet with 50% instance selection. Samples are manually selected to showcase the best quality outputs from this model.

(a) Baseline (100% of dataset)　　　(b) w\ Instance selection (50% of dataset)

Figure 8: Uncurated samples from BigGAN models trained on 256×256 resolution ImageNet. Each row is conditioned on a different class (from top): Red-breasted Merganser, Lynx, Collie, Mink, Gibbon, Barn, Castle, Drilling Platform, Promontory.

(a) Full dataset

(b) Dataset after $50\%$ instance selection

(c) Samples from GAN trained on full dataset

(d) Samples from GAN trained on $50\%$ of dataset

Figure 9: Visualization of the image manifolds for the red pandas class from (a) the full ImageNet dataset, (b) the dataset after $50\%$ instance selection, (c) samples from a GAN trained on the full dataset, and (d) samples from a GAN trained on $50\%$ of the dataset. All images are at $128 \times 128$ resolution. Manifolds are created by embedding all images into an Inceptionv3 feature space, then projecting them into 2D with UMAP [17]. All images share the same 2D embedding such that subplots are comparable. Instance selection removes images from the dataset that have unusual viewpoints or pose. Both GANs appear to cover less of the image manifold than their respective source datasets. The GAN trained on the full dataset covers some regions of the image manifold that are not covered by the model trained with instance selection, however, these regions are more likely to appear unrealistic.

## Footnotes

[5] `https://github.com/pytorch/examples/tree/master/imagenet`