[Reviews · NeurIPS 2020]

Review 1

Summary and Contributions: Summary: This paper proposes to train generative models (specifically GANs) on a carefully selected subset of a given dataset, with the goal of increasing sample quality. The authors propose to take the InceptionV3 features, fit a probabilistic model (several are considered but a simple Gaussian appears sufficient), and then select which subset to train on by scoring individual datapoints using this model. Results are presented on ImageNet using SAGANs and BigGANs.

Strengths: Strengths: This paper tackles an important problem (the extreme resource requirements of current SOTA generative models) and demonstrates a substantial speedup using a sensible and simple procedure, which looks to me to be quite easy to implement. The experimental evaluation is solid and the results are reasonably convincing.

Weaknesses: Weaknesses: My core issue with this paper’s proposal is arguably philosophical: I believe we should not be pruning our datasets in order to train generative models to better mimic the most typical samples, but should instead be training our generative models to better model the dataset, outliers and all. By eliminating outliers, this paper is effectively changing the task in a way that better plays to the current limitations of our models, which feels like a bit of a cop-out. This procedure also plays to the particularities of the classifier-feature-based metrics in a way that I would argue is slightly misleading. While I hold this objection strongly, there are two mitigating factors which will outweigh it in my scoring of this paper. The first is that there is a critical need for the reduction in resource requirements in training these models, from an ethical standpoint, from an accessibility standpoint for the broader field who do not typically have access to the very-large-scale resources originally used to train these models, and from the standpoint of increasing research velocity by enabling faster turnarounds. This work is perhaps the most effective in achieving this reduction that I have yet seen--all other approaches which claim to accelerate training do not demonstrate this improvement at an appropriate scale and must be accordingly treated with skepticism (it’s trivial to achieve improvements on CIFAR; it’s diabolically difficult to achieve it on ImageNet ), and for this reason alone I believe this is worth the attention of the community. The second mitigating factor is one which is also philosophical: I hold that reviewers should be wary of “legislating from the bench,” that is, attempting to control the direction of the field by rejecting papers whose direction they disagree with despite overall interest or momentum in that direction from the field. I think that this paper will spark discussion (and will probably draw intense ire from the likelihood modeling community) and that the discussion it sparks will have value.

Correctness: The claims, methods, and empirical methodology are largely solid; I have one note. Using Inception networks to select which datapoints one will train on definitely yields an advantage for FID/IS--you’re basically training on the images which the inception network prefers, and then evaluating using the same network. It’s worth noting that the advantage is noticeably diminished when using VGG encoders, though still present. This particularity does not necessarily detract from the main results, but should be noted.

Clarity: The paper is well written and easy to understand. I had no trouble understanding the authors at any point in the paper.

Relation to Prior Work: This paper does a good job of situating itself in the context of recent work. I would only add one suggestion for related work: “Small-GAN: Speeding Up GAN Training Using Core-sets,” https://arxiv.org/abs/1910.13540, looks relevant and should be cited.

Reproducibility: Yes

Additional Feedback: Edit post rebuttal: Thanks to the authors for their feedback. I maintain my score at a 7, but I am not willing to champion this paper as I do still feel that the direction of "tossing half the data to generate better samples" is something which is a useful technique for hobbyists wishing to do artistic projects, but not a good precedent to set in the field of generative modeling (especially given that it's already thoroughly baked into ultra-clean datasets like FFHQ or CelebA-HQ).


Review 2

Summary and Contributions: This paper purports to improve visual fidelity of GAN generation by utilizing instance selection, i.e., filtering low-likelihood samples out of the training data prior to GAN training. The authors show that this simple and general trick leads to SOTA sample quality with far lower computational requirements than the previous SOTA.

Strengths: This paper presents very concrete, sound claims which are thoroughly supported by empirical evaluation. Instance selection itself is not novel but, as far as I know, has not been hitherto examined for image generation.

Weaknesses: The main weakness of this work is that it paints an inaccurate picture of what the method accomplishes. If you were to blindly trust the paper, you would think that we had reached a new milestone in image generation. However, in reality, the method sacrifices substantial diversity to achieve this, which is not reflected in the quantitative evaluation despite being very apparent qualitatively from a small number of examples.

Correctness: The claims are correct to the extent that the authors correctly report a suite of previously-proposed evaluation metrics for image generation. However, my claim is that these previously-proposed metrics fail to capture the full picture.

Clarity: This paper is extremely well-written.

Relation to Prior Work: Yes

Reproducibility: Yes

Additional Feedback: UPDATE: I have read the author feedback. Thanks to the authors for promising to be more forthright about the diversity-quality tradeoff in the camera ready version of this paper. If this paper is indeed accepted, I would really like to see sweeping changes to the overall narrative, as well as metrics which better address diversity like the Classification Accuracy Score proposed by R4. This paper purports to improve visual fidelity of GAN generation by utilizing instance selection, i.e., filtering low-likelihood samples out of the training data prior to GAN training. Technically-speaking, this is an excellent paper. The proposed method is simple, intuitive, general, and seemingly effective. Moreover, it leads to state of the art performance on multiclass ImageNet generation with substantially lower computational requirements than the previous state-of-the-art method. The authors lay out clear claims about their method which are supported by their quantitative data. Furthermore, the effects can be seen qualitatively even on a small number of image samples (Figure 7). This work will no doubt be of substantial interest to the image generation community. On the other hand, I can’t help but disagree with the high-level takeaways the authors put forth in this work. The conclusion that I personally reach after reading this submission is that **our evaluation metrics remain broken** even after all these years of proposed alternatives to Inception Score (e.g., FID, P&R, D&C). There is a well-known tradeoff in generation between sample quality and diversity. As an extreme example, a GAN could generate one high-quality example of every ImageNet class and be assigned a high Inception Score (i.e., high quality), but this model would of course have terrible diversity by anyone’s measure. Metrics like FID, P&R, and D&C all purport to help measure diversity more so than Inception Score. However, this paper shows that by **cutting out half of the most-diverse samples in the data distribution**, we can **improve on all of these metrics** (Tables 1/3). One would expect to see that such a major hit to the model’s ability to generate diverse images (can even be judged qualitatively in Figure 7) would result in worse performance on *at least one “diversity-sensitive” metric*. Given this, there are only two conclusions I can reach: (1) these evaluation methods don’t actually capture diversity at all, or (2) we are still in a regime where we can improve these metrics purely by improving sample quality, and have yet to hit the ceiling where we would have to increase diversity to push the numbers further. Unfortunately, this isn’t really discussed in the paper, and the reader is left wondering which of these two conclusions is accurate. Overall, this method is indeed interesting and worthy of publication. In the long run, I think that the discussions and follow-up work that this paper spawns may be more important than the method itself. Apologies if this is a bit harsh but I don’t personally want to see the community relying on tricks like this a few years down the road. Accordingly, I would strongly encourage the authors to be a bit more forthcoming about the implications of this work. Namely, this method is a clever trick to produce higher-quality samples if all you care about is generating the modes of a distribution. However we have **sacrificed diversity in the process**, which is currently not reflected in the narrative of the paper.


Review 3

Summary and Contributions: One of the problems with GAN training is that the support of the learned distribution and the true data distribution do not exactly overlap. As a result, a GAN can generate spurious instances that have no support with respect to the true data distribution. Recent approaches to overcome this problem suggest post-processing techniques like rejection sampling and truncating the latent space. In contrast, this paper proposes a very novel and simple preprocessing step of instance selection for improving GAN training. The central idea is that removing from the training instances that are ‘outliers’ helps the GAN to focus on regions with stronger support. As a result, the training process is faster and yields better looking realistic examples. Specifically, the authors suggest three strategies (Gaussian Likelihood, PPCA, and kNN) for selecting instances from the training set for GAN training. The authors investigate the efficacy of the instance selection strategy on the ImageNet dataset using the modified architecture of the sota GANs. Various metrics for comparing GANs, including very recent ones such as density and coverage are used to measure the performance. The results seem to suggest that GAN versions trained using instance selection result in higher performance at a lower computational cost. [post resbuttal] I thanks the authors for providing detailed response to the queries. While some of the concerns have been adequately addressed, I still have a problem with the positioning of the paper. Clearly, a GAN learned through the proposed IS procedure does not learn the original data distribution. Thus, I am unsure if the IS strategy is sound.

Strengths: The current approaches to improving the performance of GANs focus on changing the architecture, loss functions, or post-training sampling processes. I find the proposed approach of instance selection as a preprocessing step quite interesting and refreshingly new in the context of GAN training. The experiments have been performed using some of the state of the art GAN model architectures. There is a consistent improvement in the performance all across the board. The most impressive result is a significant reduction in the dataset required for GAN training and the resulting decrease in the number of iterations (BigGAN results). It is impressive that a very simple preprocessing strategy can result in substantial improvements in the performance.

Weaknesses: I am not convinced removing instances from the training set can improve the modeling of the underlying generative process. The three instance selection techniques focus on removing instances from the tail of the distributions of the different classes. This results in redistribution of the density towards the means of the classes, meaning the GAN is learning a completely different distribution. Instance selection on the training set is problematic because the GAN can never learn the tail-enders of the distribution. However, rejection sampling post-training and truncated sampling can still generate samples that perhaps have a low likelihood. It is a different matter that these approaches suggest not to use all the generated samples. The learned GAN is not handicapped. However, this is not the case in the proposed model. I do not think IS is a good measure to evaluate the quality of the generated samples, especially for a procedure that uses instance selection. The removal of ambiguous samples from the training set ensures that the GAN also never generated ambiguous samples. Thus it is expected that the IS will be very high! I am not convinced that the results obtained for the BigGAN model are accurate. Brock et al. report an FID score of 9.6! While the authors report almost twice 12.22 Interestingly, the original BigGAN FID score is less than the BigGAN score with IS by a significant margin. Thus I am not sure if the comparison is being made across the best performing models as the authors make changes to the original BigGAN architecture. At present all the measures have been estimated on the whole training set without IS. A missing baseline that would help in contextualizing all the results is the performance on an unseen real dataset. Computing the metrics FID, Precision, Recall, Density, and Coverage on a subset of ImageNet that was not used for training will help to put a bound on the best values achievable for the dataset. This would in turn help to understand the significance of the changes to various measures. For example, the change in the recall measure is around 0.04. Is this significant? I would also have liked to see some analysis of the instance selection strategies. At present, there is only a single table comparing the performances. More discussion on the suitability of the strategies will be helpful to the readers. Finally, there is a wide range of classes in ImageNet. The images within a class and across classes are very diverse. It might be easier to detect such noisy examples through the sample selection strategies. It would be helpful if the proposed method can be tested on a dataset like CelebA HD, which has more intra-class diversity (subtle differences between instances).

Correctness: There are some concerns with the experimental setup. These have been highlighted in the review.

Clarity: The paper is very well written and is easy to follow.

Relation to Prior Work: The authors discuss related work in detail and clearly articulate the differences between the proposed work and existing literature.

Reproducibility: Yes

Additional Feedback:


Review 4

Summary and Contributions: The paper proposes to simplify GAN training by discarding training samples that belong to the low-density regions of the image manifold. The authors confirm that such simplification of training set results in more efficient training and more powerful models.

Strengths: -- The problem in question is definitely important since training of the state-of-the-art GANs requires too much computational resources. -- The paper is exceptionally well-written. -- The proposed approach is very handy and simple, which is a virtue. -- The experiments are properly designed and technically sound. I also appreciate the sections with preliminary analysis and demonstrative qualitative examples.

Weaknesses: -- I am not sure if the proposed simplification of the training set would not result in the generative model that produces less useful synthetic data compared to the model trained on the full set. The reported metrics (FID, Precision, Coverage, Density, Recall) are only proxies and more downstream-oriented metrics like Classification Accuracy Score (NeurIPS'2019) should be reported. Since the authors aim to generate labeled Imagenet images, CAS can be computed and compared to the numbers from the literature. -- The proposed approach heavily relies on embeddings from the pretrained classifier, which is available for ImageNet, but can be absent for GANs in completely unsupervised setups. For example, how can one use the proposed techniques for WaveGAN to generate spectrograms? Overall, my concern is that often GANs are expected to operate in a completely unsupervised setups and it is not clear if the state-of-the-art unsupervised embeddings (e.g. SimCLR) are good representations for the proposed technique. -- The paper reports profit in terms of all metrics only for relatively simple models (SAGAN), but reports only IS/FID for the state-of-the-art BigGAN. Are there benefits in terms of Density/Coverage/Precision/Recall for BigGAN? I am asking since simplifying training sets makes more sense for simpler models, but at the end of the day our goal is to generate high-res images.

Correctness: Yes.

Clarity: Yes.

Relation to Prior Work: Yes.

Reproducibility: Yes

Additional Feedback: While I like the overall premise and approach taken in this paper, I have several concerns regarding the method and its evaluation that have led to my score. I look forward to reading the authors' perspective during the rebuttal phase. ----------- AFTER REBUTTAL --------- I have read the rebuttal and I appreciate that the authors have addressed my concerns. After rebuttal, I keep my original recommendation (5: Marginally below acceptance threshold). (1) As I requested, the authors report Classification Accuracy Score[1], which I consider a more reliable measure of diversity/usefulness of synthetic data. As expected, IS harms CAS significantly, moreover, all the numbers are much lower compared to [1]. This means that for downstream ML tasks, IS is probably detrimental. (2) The authors claim that their goal is to produce realistic data at the cost of diversity, which is closer to graphics-oriented research on GANs. However, common benchmarks in this line of research are less diverse compared to Imagenet, e.g. FFHQ face images are aligned and cropped, which is also a form of dataset curation. Therefore, IS will be probably less useful here, as confirmed by the rebuttal's results on CelebA. Given (1)-(2), I am not convinced that IS is a right tool for both industry and research. The valuable part of this paper is that its experiments indicate the current measures (IS/FID/P/R) are insufficient. But for a general reader it can be unclear from the current text, since it does not criticize IS/FID/P/R at all. [1] Classification Accuracy Score for Conditional Generative Models, NeurIPS' 2019

[Author Response · NeurIPS 2020]

We would like to thank all reviewers for their time and consideration in reviewing our paper. We were impressed by the quality of the feedback and insights, and appreciate the reviewers' recognition of the benefits afforded by our work. We include here some comments from the reviewers. R1: "This work is perhaps the most effective in achieving [training time] reduction that I have yet seen". "This paper will spark discussion... and the discussion it sparks will have value". R2: "This work will no doubt be of substantial interest to the image generation community". R3: "Quite interesting and refreshingly new". "It is impressive that a very simple preprocessing strategy can result in substantial improvements in the performance". R4: "Exceptionally well-written". "Very handy and simple, which is a virtue". The reviewers also raised some questions to which we respond below. In the rebuttal, IS stands for Instance Selection (**not Inception Score**), while P, R, C and D stand for Precision, Recall, Density and Coverage metrics.

**R1** Progress should be made with model adaptation, not dataset adaptation. Indeed, this is a philosophical question. Does it make sense to separate the model and the data if we can benefit downstream tasks by considering them jointly? We look forward to the discussions that our work will generate in this regard. Inception embedding yields advantages under certain evaluation metrics. We agree. We will add a note to clarify this. Small-GAN. We will cite this as related work in the camera-ready.

**R2** Effectiveness of diversity-sensitive evaluation metrics. Of the metrics considered, two specifically target diversity captured by the model: R and C. R can be thought of as a measure of *unconstrained diversity*, where we care less about image quality and more about the spread of the samples. C, by definition, reflects *realism-constrained diversity*, where we are interested in modeling diversity among realistic looking model samples. In all of our experiments we observe that the application of IS does reduce R, which is expected given that the diversity of the training set is reduced. With C we observe an initial increase in performance, but when too much of the original dataset is removed it too begins to drop. We see this in Figure 3 in the C subplot, where retention ratio 20 hovers at a lower C value than retention ratio 30 before its collapse. This example demonstrates that we need to lose a lot of the diversity in the dataset before it becomes more of a problem than the gains from better image quality. This would also explain why FID improves when IS is applied, despite reduced diversity in the training set. We believe this follows the conclusion that "we are still in a regime where we can improve these metrics purely by improving sample quality, and have yet to hit the ceiling where we would have to increase diversity to push the numbers further". We will add discussion of this to the paper. Not enough emphasis on diversity trade-off. In the camera-ready we will emphasize that IS is a tool for trading diversity captured in the dataset for image fidelity, and that this trade-off is controlled by the IS retention rate.

**R3** Post IS the model cannot generate tails of the distribution. While this is true, we have shown experimentally that for current GANs these outliers might result in a worse fit of the true data distribution. This gives practitioners a tool that they can choose to use depending on whether they prefer image fidelity, or the chance to generate distribution tails. Inception Score is not a good measure. We agree. We include it for completeness since it is a standard metric to report, despite its many shortcomings. BigGAN does not match reported results. We train our BigGAN models using the official PyTorch code on a GPU server. We change a single hyperparameter, the channel multiplier, from 96 to 64 to lower the memory requirements and speed up training, which is why our baseline does not match the FID of 9.77 achieved by the official implementation. As a note of interest, after the submission deadline we found that training IS models with a smaller batch size of 256 further improved performance (likely a regularization effect), achieving an FID of 9.61 in less than 4 days. Thus, we can outperform the official full capacity BigGAN model while using half the trainable parameters, and still train in $\frac{1}{3}$ the time. Baseline evaluation metrics on real data. This is a great idea, we will include it in the camera-ready! CelebA HD. Datasets with less diversity already have high manifold density. As such, we expect the impact of IS to be less pronounced in these cases. CelebA HD has already been heavily pre-processed with face alignment (which might be considered a form of IS), so we would not expect much additional benefit from IS on this dataset. We train several SNGAN models (baseline and 95% IS) on CelebA HD and observe average improvements in P (0.83 to 0.85) and D (1.09 to 1.27) and impairments to FID (11.92 to 13.25) and R (0.41 to 0.35). C is unchanged (0.84 for both). Thus IS still achieves the desired effect of trading diversity for visual fidelity, but the trade-off is less appealing for well curated datasets. We note that these results were achieved using an ImageNet pretrained embedding, so better performance may still be achieved by using a domain-specific face embedding.

**R4** Classification Accuracy Score (CAS). Generally, classifiers prefer diverse data over realistic data, which is why data augmentation techniques such as Mixup and CutMix perform well even though they produce unrealistic images. Since the goal of IS is to improve image quality by reducing diversity, we do not expect it to be helpful for data augmentation. We measure CAS using a ResNet18 architecture for quicker training. Our baseline $128 \times 128$ BigGAN model achieves a Top-1 accuracy of 8.69 and Top-5 of 18.51, while our IS model reaches a slightly lower Top-1 accuracy of 6.31 and Top-5 of 16.04. This falls in line with our expectations. Unsupervised embeddings. We trained a model using a ResNet50 SwAV embedding (Caron et al., 2020), which was recently released and shown to outperform SimCLR. This model achieves an FID of 16.65 after 200k iterations (compared to the baseline of 21.66), demonstrating that unsupervised embeddings can be used for IS. Learning useful unsupervised embeddings for domains outside of images is still an open problem, but we see this as an exciting direction for future work. PRDC for BigGAN experiments. We computed P&R and D&C for our BigGAN experiments and observe a similar trend to our SAGAN experiments: P, D, and C increase over the baseline, while R decreases. Baseline/IS: P=0.76/0.88, R=0.66/0.50, D=0.90/1.34, C=0.84/0.90.

[Meta-Review · NeurIPS 2020]

Reviewers were almost unanimous in voting to accept this paper, and I think overturning the reviewer decisions should be done very cautiously, so I will recommend acceptance here. However, I have a serious problem with this paper that I really hope that authors will address: Your results are interesting, but you are wrong about why they are interesting! The reviewers have discussed this at length, but just to summarize, you have not shown a technique for actually making GANs better at doing the thing that people want them to do, you have instead shown why the way the people evaluate GANs is all wrong. I think it would be a real disservice to the community (and cause a bunch of unnecessary confusion) to not change the narrative in the paper to reflect this. If you don't make this change, I will be very sad, and I bet the reviewers will be sad also, and besides, from a selfish perspective, I think the paper will be better received if you make this change anyway.